# Impaired Lung Function and Lung Cancer Incidence: A Nationwide Population-Based Cohort Study

**DOI:** 10.3390/jcm11041077

**Published:** 2022-02-18

**Authors:** Hye Seon Kang, Yong-Moon Park, Seung-Hyun Ko, Seung Hoon Kim, Shin Young Kim, Chi Hong Kim, Kyungdo Han, Sung Kyoung Kim

**Affiliations:** 1Division of Pulmonary, Allergy and Critical Care Medicine, Department of Internal Medicine, Bucheon St. Mary’s Hospital, College of Medicine, The Catholic University of Korea, Bucheon 14647, Korea; ihbe3805@gmail.com; 2Department of Epidemiology, Fay W. Boozman College of Public Health, University of Arkansas for Medical Sciences, Little Rock, AR 72205, USA; ypark@uams.edu; 3Division of Endocrinology and Metabolism, Department of Internal Medicine, St. Vincent’s Hospital, College of Medicine, The Catholic University of Korea, Suwon 16247, Korea; kosh@catholic.ac.kr; 4Division of Pulmonary and Critical Care Medicine, Department of Internal Medicine, St. Vincent’s Hospital, College of Medicine, The Catholic University of Korea, Suwon 16247, Korea; hotboat@catholic.ac.kr (S.H.K.); newzero82@catholic.ac.kr (S.Y.K.); chihongk@catholic.ac.kr (C.H.K.); 5Department of Statistics and Actuarial Science, Soongsil University, Seoul 06978, Korea

**Keywords:** lung cancer, pulmonary function test, lung function impairment

## Abstract

Background: It is unclear whether the presence of minimal lung function impairment is an independent risk factor for the development of lung cancer in general populations. Methods: We conducted a population-based cohort study using nationally representative data from the Korean National Health and Nutrition Examination Survey and the Korean National Health Insurance Service. Results: Of 20,553 participants, 169 were diagnosed with lung cancer during follow-up (median, 6.5 years). Participants with obstructive lung function impairment had increased risk of lung cancer (aHR: 2.51; 95% CI: 1.729–3.629) compared with those with normal lung function. The lower was the quartile or decile of forced expiratory volume in one second (FEV1) or the FEV1/forced vital capacity (FVC) ratio, the significantly higher was the incidence rate of lung cancer (*p* for trend < 0.0001). With FEV1 values in the lowest quartile (Q4), the incidence of lung cancer was significantly increased regardless of FVC (FEV1 Q4 and FVC values in the higher three quartiles Q1–3: aHR 1.754; 95% CI 1.084–2.847, FEV1 Q4 and FVC Q4: aHR 1.889; 95% CI 1.331–2.681). Conclusion: Our findings suggest that minimal lung function impairment, as expressed by lower FEV1 or FEV1/FVC value, may be associated with increased risk of lung cancer

## 1. Introduction

Lung cancer is the leading cause of death from cancer worldwide [1]. Approximately 70% of patients have advanced disease at the time of diagnosis, and only 15% of patients with lung cancer are alive five years after diagnosis [2]. Thus, early detection of lung cancer is very important; for this purpose, low-dose computed tomography (LDCT) screening is performed in high-risk groups for lung cancer [3].

Tobacco smoking is the most important risk factor for lung cancer, although exposures to other agents such as radon, asbestos, and environmental tobacco smoke (ETS) also are involved [4]. In addition, chronic obstructive pulmonary disease (COPD) and other smoking-related diseases have been found to be associated with higher rate of lung cancer in several studies [5,6,7,8]. Additionally, obstructive lung function impairment based on forced expiratory volume in one second (FEV1) has been reported to be associated with lung cancer risk in smokers or groups of men with other characteristics [9,10,11,12,13]. However, it is unclear whether the presence of minimal lung function impairment can be considered an independent risk factor for the development of lung cancer in general populations.

The pulmonary function test (PFT) is a cost-effective, easy, and fast tool for diagnosing lung function impairment. With this tool, the identification of individuals with higher lung cancer risk on the basis of lung function decline can be used as a determining parameter and establish cut-off values for the prediction and early detection of lung cancer [9].

The aim of the present study is to identify an association between lung function and lung cancer development in a large, nationwide database using linkages between the 2010–2016 Korea National Health and Nutrition Examination Survey (KNHANES) and the National Health Insurance Service (NHIS) claims database in the Korean population.

## 2. Materials and Methods

### 2.1. Database and Study Population

Since 1998, the KNHANES has been regularly conducted under the leadership of the Korea Disease Control and Prevention Agency to monitor the general health and nutritional status of the civilian, noninstitutionalized Korean population [14]. Korea’s NHIS is a social insurance payment system that covers about 97% of the Korean population. The NHIS data include all national routine health exam and claims data. Claims data include drug prescriptions, diagnostic codes for the International Classification of Disease-10 (ICD-10) disease coding system, and detailed treatment information for all patients [15]. The present study used KNHANES data collected between 2008 and 2016. Of 40,279 KNHANES participants, adults over 40 years of age who had undergone spirometry tests were included in our analysis. We excluded subjects with missing data and those previously diagnosed with lung or any other cancer before 1 January 2008. To assure the primary endpoint of newly diagnosed lung cancer, we established a washout time of more than one year. Eligible subjects selected from the KNHANES database were merged with those from the NHIS database, producing a cohort dataset. To evaluate newly diagnosed lung cancer, we used these cohort data from 2008 with clinical follow-up through 31 December 2016.

The Institutional Review Board of The Catholic University of Korea (IRB no: HC21ZISI0063) approved this study. The study was conducted in compliance with the Declaration of Helsinki.

### 2.2. Clinical and Laboratory Measurements

Details of the KNHANES framework regarding the content of health surveys, standardized physical examinations, laboratory tests, and definitions of risk factors have been described previously [15]. Among participants herein, specialists performed physical examinations according to standardized methods. Body mass index (BMI) was calculated as participant body weight in kilograms divided by the square of height in meters. Waist circumference was measured at the midpoint between the lowest rib and the anterior iliac crest of participants in the standing position. Health-related behavior surveys included well-established questions to determine demographic and socioeconomic characteristics of the population. Smoking status was divided into three categories: nonsmoker, ex-smoker, or current smoker. Alcohol consumption was assessed based on the average number of alcoholic beverages and frequency of drinking. Heavy drinkers were defined as subjects who drank more than 30 g/day, while subjects drinking less than 30 g/day were classified as mild to moderate drinkers [16]. Moderate physical activity was defined as walking at least 150 min per week [16]. Household income was divided into quartile groups of lowest, lower middle, higher middle, and highest. A high level of education was defined as completion of high school or above.

Diabetes mellitus (DM) was defined as a fasting glucose level ≥ 126 mg/dL, current use of anti-diabetic medications, or a self-reported physician diagnosis [17]. Hypertension was defined as systolic blood pressure ≥ 140 mmHg or diastolic blood pressure ≥ 90 mmHg, current use of anti-hypertensive medications, or a self-reported physician diagnosis [18]. Hypercholesterolemia was defined as total cholesterol ≥ 240 mg/dL, current use of cholesterol-lowering medications, or a self-reported physician diagnosis. A total of 18 Blood samples were collected following overnight fasting by participants.

### 2.3. Spirometry

Spirometry is one of the tools used to evaluate and monitor health status in general population provided by KNHANES. Spirometry was performed by four technicians, each of whom underwent two education sessions for lung function testing and quality control. Trained technicians measured FEV1, forced vital capacity (FVC), and the FEV1/FVC ratio using a dry rolling seal spirometer (model 2130; Sensor Medics, Yorba Linda, CA, USA) and the American Thoracic Society/European Respiratory Society criteria for standardization of lung function tests [19]. All spirometry values were described in terms of pre-bronchodilator results [20]. Normal predictive values were derived considering healthy subject age, sex, height, and ethnicity from a large population study [21]. Analyses were performed only on data that met the following criteria: (i) two acceptable spirometry curves showing correct start of the test and expiration for at least six seconds and (ii) the greatest difference between two measurements of FEV1 or FVC < 150 mL. Spirometry results were classified into three groups of normal, non-obstructive, and obstructive lung function impairment. Participants with FEV1/FVC ≥ 0.7 and FVC ≥ 80% of the normal predicted value were considered normal. Non-obstructive pattern was defined as FEV1/FVC ≥ 0.7 and FVC < 80% predicted, and obstructive pattern was defined as FEV1/FVC < 0.7 [22].

### 2.4. Clinical Outcomes

The primary outcome was newly diagnosed lung cancer during the established follow-up period. Since 2005, the Korean government has implemented policies to expand the benefit coverage of NHIS to provide financial protection against life-changing and catastrophic diseases such as cancer. This NHIS program reimburses 95% of the costs of catastrophic diseases such as cancer. When patients with lung cancer are registered in this system, they are assigned a special code (V code). We identified patients with lung cancer using both ICD-10 (C33, C34) and V codes (V193), following protocols established in a previous study [23].

### 2.5. Statistical Analysis

Summary statistics are expressed as means and standard deviations for continuous variables and as numbers and percentages for categorical variables. Continuous variables were compared using Student’s *t*-test or analysis of variance, as appropriate. Categorical variables were compared using Chi-square test. The incidence rate of lung cancer was calculated by dividing the number of lung cancer patients by the sum of the follow-up duration, presented as the rate per 1000 person–years. Participants were followed until the first diagnosis of lung cancer or censoring by death or date of 31 December 2016. The survival and disease-free probability of incident lung cancer according to the lung function was calculated using the Kaplan–Meier method and the log-rank test was conducted to analyze differences among the groups. Cox proportional-hazard models were used to estimate hazard ratios (HRs) and 95% confidence intervals (CIs) for lung cancer incidence. The provided *p* values are two-sided, with the level of significance at 0.05. Multivariable regression models were constructed with non-adjustment (model 1); including age, sex, BMI, smoking, alcohol consumption, household income, and exercise (model 2); and including the variables in model 2 plus the presence of DM, hypertension, and hypercholesterolemia (model 3). All statistical analyses were performed using SAS version 9.4 (SAS Institute, Cary, NC, USA).

## 3. Results

We identified 40,279 participants by linking KNHANES and NHIS datasets from 2008 to 2016. Of these, 4772 participants under the age of 40, 11,479 participants with missing PFT records, 1299 participants with history of malignancy, and 2176 participants with missing data were excluded. Finally, 20,553 participants were analyzed (Figure 1).

### 3.1. Baseline Characteristics of Study Participants

Table 1 details the baseline characteristics of participants included in this study. Among the study participants, the proportions of obstructive and non-obstructive lung function impairment were 13.1% and 10.2%, respectively. The proportions of older age, male, current smoker, heavy alcohol consumption, less educated, lowest quartile of income, DM, and hypertension were significantly higher in subjects with obstructive or non-obstructive lung function impairment than in those with normal lung function. Subjects with non-obstructive lung function impairment had a higher mean BMI and waist circumference than subjects with obstructive lung function impairment or normal lung function.

Of the study participants, 169 (0.82%) were diagnosed with lung cancer during the follow-up period (Table 2). The median duration of follow-up was 6.5 (interquartile range 4.5–8.5) years. Subjects with lung cancer had significantly higher percentages of older age, male, smoking history (ex- or current smoker), less educated, lowest quartile of income, DM, and hypertension. In PFT, mean FVC (88.97% vs. 92.71%, *p* < 0.0001), mean FEV1 (83.81% vs. 92.25%, *p* < 0.0001), and mean FEV1/FVC (0.7 vs. 0.78, *p* < 0.0001) were lower in the lung cancer group than in the control group.

### 3.2. Risk of Lung Cancer According to Lung Function Impairment Pattern

Table 3 shows adjusted hazard ratios (HRs) and 95% CIs for the association between lung function and the risk of incident lung cancer. We grouped participants into three groups (normal, obstructive, and non-obstructive lung function impairment) based on PFT. In comparison to participants with normal PFT results, the unadjusted HR was 5.817 (4.222–8.014) in the obstructive lung function impairment group and 1.837 (1.112–3.033) in the non-obstructive lung function impairment group. After adjusting for age, sex, BMI, income, smoking, alcohol consumption, and moderate physical activity (model 2), the adjusted HR (95% CI) was 2.518 (1.739–3.648) in the obstructive lung function impairment group and 1.296 (0.776–2.166) in the non-obstructive lung function impairment group in comparison to the normal lung function group. When DM, hypertension, and hypercholesterolemia were additionally controlled (model 3), the adjusted HR (95% CI) was 2.505 (1.729–3.629) in the obstructive lung function impairment group and 1.273 (0.761–2.129) in the non-obstructive lung function impairment group in comparison to the group with normal lung function.

### 3.3. Risk of Lung Cancer According to Lung Function Quartile

Table 4 shows adjusted HRs and 95% CIs for the risk of incident lung cancer according to lung function quartile. For FEV1 quartiles, the unadjusted HR (95% CI) was 3.528 (2.217–5.613) in the lowest quartile (Q4) of FEV1 in comparison to the highest quartile (Q1). When age, sex, BMI, income, smoking, alcohol consumption, and moderate physical activity were controlled (model 2), the adjusted HR (95% CI) was 2.854 (1.776–4.589) in the lowest quartile of FEV1 in comparison to the highest quartile. With additional adjustment for DM, hypertension, and hypercholesterolemia (model 3), the adjusted HR (95% CI) was 2.845 (1.769–4.575) in the lowest quartile of FEV1 in comparison to the highest quartile.

For FVC quartiles, the unadjusted HR was 2.141 (1.403–3.265) in the lowest quartile of FVC in comparison to the highest quartile. However, lung cancer incidence was not different between the highest quartile and the rest of the quartiles with model 2 or model 3.

For FEV1/FVC, adjusted HRs (95% CIs) in the lowest quartile of FEV1/FVC, in comparison to the highest quartile, according to model 1, model 2, and model 3, were 7.696 (4.395–13.476), 2.886 (1.571–5.301), and 2.891 (1.573–5.312), respectively.

### 3.4. Risk of Lung Cancer According to Lung Function Decile

The cumulative incidence function curves of lung cancer according to lung function decile are plotted in Figure 2. Patients in the lower deciles of FEV1, FVC, and FEV1/FVC were at a higher cumulative incidence of lung cancer (*p* < 0.0001). Table 5 shows adjusted HRs and 95% CIs for the risk of incident lung cancer according to lung function decile. For FEV1 deciles, the unadjusted HR (95% CI) was 4.118 (2.13–7.961) in the lowest decile (D10) in comparison to the highest decile (D1). When age, sex, BMI, income, smoking, alcohol consumption, and moderate physical activity were controlled (model 2), the adjusted HR (95% CI) was 3.269 (1.67–6.397) in the lowest decile of FEV1 in comparison to the highest decile. With additional adjustment for DM, hypertension, and hypercholesterolemia (model 3), the adjusted HR (95% CI) was 3.277 (1.674–6.416) in the lowest decile of FEV1 in comparison to the highest decile.

For FVC deciles, the unadjusted HR was 2.961 (1.493–5.875) in the lowest decile compared to the highest decile. However, lung cancer incidence was not different between the lowest and the rest of the deciles with model 2 or model 3.

For FEV1/FVC, adjusted HRs (95% CIs) in the lowest decile of FEV1/FVC in comparison to the highest deciles according to model 1, model 2, and model 3 were 11.742 (5.078–27.151), 3.579 (1.462–8.761), and 3.593 (1.467–8.801), respectively.

### 3.5. Risk of Lung Cancer According to Continuous Variables of Lung Function

We further analyzed the risk of lung cancer according to increase in lung function based on continuous variables of FEV1, FVC and FEV1/FVC values. For FEV1, the adjusted HRs for every 1% increase in FEV1 was 0.974 (*p* < 0.0001) in Model 2 and Model 3, respectively. Similarly, the adjusted HRs for every 0.01 increase in FEV1/FVC was 0.95 (*p* < 0.0001) in Model 2 and Model 3. The unadjusted HR for every 1% increase in FVC was 0.973 (*p* < 0.0001), but there was no statistically significant difference according to increase in FVC after adjustment (Table 6).

### 3.6. Risk of Lung Cancer According to Quartile Combination of FEV1 and FVC

To determine which of FEV1 or FVC has a greater impact on lung cancer incidence, the risk of lung cancer by quartile combination of FEV1 and FVC was analyzed (higher three quartiles: Q1–3 vs. lowest quartile: Q4) (Table 7). A group with the higher three quartiles (Q1–Q3) for both FEV1 and FVC served as the reference. When FEV1 values were in the lowest quartile (Q4), the incidence of lung cancer was significantly increased regardless of FVC. In particular, the incidence of lung cancer was highest in the group with the lowest quartiles for both FEV1 and FVC (FEV1 Q4 and FVC Q4: aHR 1.889; 95% CI 1.331–2.681). However, when only FVC values were in the lowest quartile, there was no significant difference in the incidence rate of lung cancer (FEV1 Q1–3 and FVC Q4: aHR 0.672; 95% CI 0.334–1.351).

## 4. Discussion

In this study using nationally representative data in the Korean population, we observed that decreased lung function was associated with increased risk of lung cancer after adjusting for various confounding factors. Individuals with obstructive or non-obstructive lung function impairment showed a higher risk of lung cancer compared with those with normal lung function. Further, we found that those with lower quartiles or deciles of FEV1 or FEV1/FVC had a higher risk of lung cancer.

The relationship between COPD and lung cancer has been recognized. In a cohort of male construction workers, a high rate of lung cancer was observed in a COPD group relative to a group with normal lung function [6]. Additionally, the presence of COPD has been associated with a higher risk for lung cancer incidence in adult general populations in the US and UK [7,8]. In a nationwide population-based cohort, COPD was an independent risk factor for development of lung cancer regardless of smoking status [5].

Several studies have suggested that airway obstruction, based on FEV1 reduction, increases lung cancer risk. In a community-based cohort of Japanese-American men, the percentage of predicted FEV1 was inversely associated with risk of lung cancer [10]. Additionally, FEV1 was inversely associated with risk of lung cancer among former and current smokers but not in never-smokers [9,11,12]. Further, a strong linear relationship was observed between increasing severity of airflow limitation and risk of lung cancer in heavy smokers [13]. In never-smokers, impaired lung function in the risk prediction model for lung cancer showed a limited improvement in predictive performance [24]. However, it is unclear whether the presence of minimal lung function impairment should be considered an independent risk factor for the development of lung cancer in general populations.

One difference between our study and the existing research is that we separated the evaluation of obstructive and non-obstructive lung function impairment. In addition, pulmonary function parameters were subdivided into quartiles, deciles or change of continuous variables, and the relationship between lung function and lung cancer development was investigated by group of or change in lung function values. We showed that minimal and moderate obstructive lung function impairment confers an increased risk of lung cancer development in the general population after adjusting for confounding factors.

One important clinical application of our study is the use of spirometry to better target CT screening for early detection of lung cancer. In a previous similar approach, inclusion of spirometric criteria for CT screening eligibility resulted in an increase in lung cancer detection of 6.8%, which is higher than in other studies where screening populations were identified based on age and smoking history [25,26]. Lung cancer screening in individuals with lung function impairment is not recommended by the US Preventive Services Task Force [27]. Similarly, the highest-risk group subject to screening comprises people between 54 and 74 years of age, who were recorded as current smokers with a smoking history of 30 pack years or more in the health checkup or smoking cessation treatment support project questionnaire in the previous year in Korea. The importance of lung cancer screening is emphasized by the mortality reduction seen in the recently published, large, randomized, NELSON screening trial [3].

Smoking exposure is an important prerequisite for lung function impairment. Additionally, there is sufficient evidence to establish a causal association between smoking and lung cancer [28]. However, the proportion of never-smoker lung cancer patients are increasing [29]. Further, the contribution of smoking in comparison to the variance in ventilatory function is modest and much less meaningful than genetic factors in most lung cancer [30,31]. The higher susceptibility of the lungs to cancer due to smoking is due to the combined effects of inflammation and aberrant repair [32]. Lung function decline and COPD also are caused by indoor air pollutants, poorly controlled chronic asthma, occupational exposures to dusts, poor socioeconomic status, malnutrition, childhood respiratory infections, and formerly treated pulmonary tuberculosis even without a smoking history [33,34]. In our study, minimal lung function impairment was one of the dependent risk factors for lung cancer risk after adjusting for confounding factors including smoking history. Even though smoking is one of attributable factor for lung cancer risk, our study suggests that minimal lung function impairment can be a dependent risk factor for lung cancer development and has clinical implications for lung cancer screening in the general population.

Some proposed mechanisms for poor lung function and lung cancer risk include the impaired pulmonary clearance of inhaled carcinogens and inflammation-induced production of genotoxic reactive oxygen species [35]. In addition, chronic inflammation caused by accumulation of mucous exudates in the lumen, leading to the remodeling and thickening of bronchiolar walls associated with impaired tissue repair, could result in the production of several growth factors and growth of sporadically transformed cells [36,37].

Although it is clear that smoking plays an important role in the development of lung cancer and lung function decline, lung function deterioration not associated with smoking also contributes. Accordingly, if lung function parameters are added to the selection of subjects for lung cancer screening (currently based on smoking history and age), the specificity over sensitivity of lung cancer screening can be maximized to result in a more favorable trade-off between the harms and benefits of LDCT screening. The results of our study have the potential to be used as basic data for selecting high-risk groups for lung cancer screening based on lung function parameters.

A limitation of this study is that detailed history of ETS, e-cigarettes, and exposure to occupational dusts, which are associated with lung function impairment and/or lung cancer risk, was not included in the analysis. Additionally, other factors, such as drugs for airway disease and combined emphysema involved in lung cancer development, were not analyzed. We also did not consider smoking amounts, one of the confounding factors for lung cancer development, but considered current smoking status as an adjustment factor. Further, cell types and lung cancer stage were not investigated according to lung function impairment due to the limitations of data collection.

In conclusion, the findings from this nationally representative, Korean population-based large cohort study support the hypothesis that lower FEV1 or FEV1/FVC are associated with lung cancer incidence. The present study indicates the role of PFT as a noninvasive, affordable, and fast tool in screening for optimal candidates for the early detection of lung cancer.

## 5. Interpretation

It is unclear whether the presence of minimal lung function impairment is an independent risk factor for the development of lung cancer in general populations. We showed that minimal lung function impairment is significantly associated with increased incidence of lung cancer in general population. The results of our study may serve as basic data for determining which subjects could be considered for screening for the early detection of lung cancer.

## Figures and Tables

**Figure 1 jcm-11-01077-f001:**
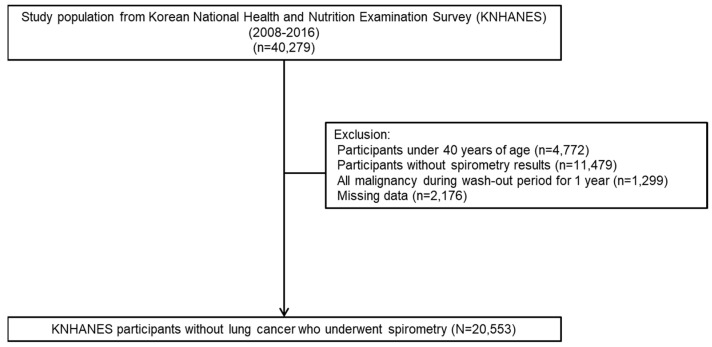
Flow chart of the study population.

**Figure 2 jcm-11-01077-f002:**
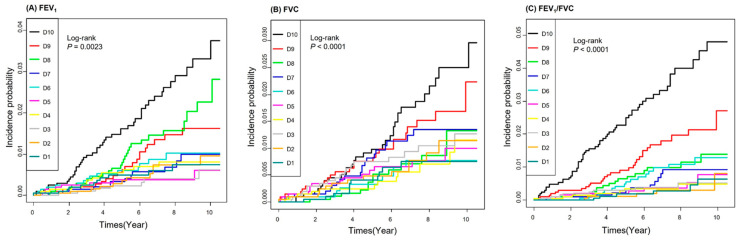
Cumulative incidence function curves of lung cancer according to lung function decile. (**A**) FEV1, (**B**) FVC, (**C**) FEV1/FVC.

**Table 1 jcm-11-01077-t001:** Baseline characteristics of participants according to lung function pattern.

	Lung Function			*p*-Value
	Normal (*n* = 15,775)	Obstructive (*n* = 2686)	Non-Obstructive (*n* = 2092)	
Age	55.68 ± 10.06	65.55 ± 9.54	60.90 ± 10.64	<0.0001
Age, ≥65 year	3333 (21.13)	1568 (58.38)	828 (39.58)	<0.0001
Sex				<0.0001
Male	6184 (39.20)	1975 (73.53)	1001 (47.85)	
Smoking				<0.0001
Non	10252 (64.99)	849 (31.61)	1219 (58.27)	
Ex	2910 (18.45)	1014 (37.75)	477 (22.80)	
Current	2613 (16.56)	823 (30.64)	396 (18.93)	
Alcohol consumption				<0.0001
Non	4516 (28.63)	853 (31.76)	753 (35.99)	
Mild	10017 (63.50)	1533 (57.07)	1166 (55.74)	
Heavy	1242 (7.87)	300 (11.17)	173 (8.27)	
High Education (≥high school %)	9359 (59.33)	1114 (41.47)	1004 (47.99)	<0.0001
Household Income (lowest quartile %)	2618 (16.60)	847 (31.53)	514 (24.57)	<0.0001
Moderate physical activity	6163 (39.07)	1146 (42.67)	772 (36.90)	0.0001
Diabetes Mellitus	1693 (10.73)	484 (18.02)	493 (23.57)	<0.0001
Hypertension	5514 (34.95)	1328 (49.44)	1091 (52.15)	<0.0001
Hypercholesterolemia	3088 (19.58)	509 (18.95)	501 (23.95)	<0.0001
BMI	24.16 ± 2.86	23.69 ± 2.75	25.32 ± 3.23	<0.0001
Waist circumference	77.62 ± 8.98	79.76 ± 8.82	82.00 ± 9.54	<0.0001
Glucose	100.25 ± 22.31	103.47 ± 22.75	108.34 ± 30.58	<0.0001
SBP	120.83 ± 16.4	125.55 ± 16.72	126.72 ± 17.34	<0.0001
DBP	77.72 ± 10.17	76.21 ± 10.55	78.06 ± 10.48	<0.0001
Total Cholesterol	195.81 ± 35.82	188.57 ± 36.66	192.82 ± 37.66	<0.0001
HDL-cholesterol	49.45 ± 11.8	46.87 ± 11.62	46.45 ± 11.38	<0.0001
FEV1 (%)	96.46 ± 10.32	78.35 ± 15.33	77.64 ± 7.88	<0.0001
FVC (%)	95.57 ± 9.38	90.33 ± 14.03	73.94 ± 5.83	<0.0001
FEV1/FVC	0.80 ± 0.05	0.64 ± 0.07	0.80 ± 0.05	<0.0001
AST *	21.54 (3.06–21.65)	22.74 (3.11–23.02)	23.07 (3.12–23.41)	<0.0001
ALT *	19.07 (2.94–19.22)	19.45 (2.95–19.79)	21.47 (3.04–21.95)	<0.0001
TG *	119.03 (4.77–120.12)	126.34 (4.82–129.01)	133.25 (4.87–136.52)	<0.0001

Values are number (%) or mean ± standard deviation. * Geographic mean (95% confidence interval). Abbreviation: ALT, alanine aminotransferase; AST, aspartate aminotransferase; BMI, body mass index; DBP, diastolic blood pressure; FEV1, forced expiratory volume in one second; FVC, forced vital capacity; HDL, high-density lipoprotein; SBP, systolic blood pressure; TG, triglycerides.

**Table 2 jcm-11-01077-t002:** Baseline characteristics of participants with and without lung cancer.

	Lung Cancer		*p* Value
	Yes (*n* = 169)	No (*n* = 20,384)	
Age, years	65.53 ± 9.07	57.43 ± 10.63	<0.0001
Age, ≥65 year	94 (55.62)	5635 (27.64)	<0.0001
Sex, male	106 (62.72)	9054 (44.42)	<0.0001
Smoking			<0.0001
Non	64 (37.87)	12,256 (60.13)	
Ex	50 (29.59)	4351 (21.35)	
Current	55 (32.54)	3777 (18.53)	
Alcohol consumption			0.1101
Non	58 (34.32)	6064 (29.75)	
Mild	92 (54.44)	12,624 (61.93)	
Heavy	19 (11.24)	1696 (8.32)	
High Education (≥high school %)	58 (34.32)	11,419 (56.02)	<0.0001
Household Income (lowest quartile %)	55 (32.54)	3924 (19.25)	<0.0001
Moderate physical activity	73 (43.20)	8008 (39.29)	0.3001
Diabetes Mellitus	35 (20.71)	2635 (12.93)	0.0027
Hypertension	79 (46.75)	7854 (38.53)	0.0289
Hypercholesterolemia	26 (15.38)	4072 (19.98)	0.1368
BMI	23.60 ± 2.87	24.22 ± 2.92	0.0059
Waist circumference	78.93 ± 8.73	78.34 ± 9.13	0.3971
Glucose	104.67 ± 24.28	101.47 ± 23.47	0.0773
SBP	127.34 ± 18.14	122.00 ± 16.67	<0.0001
DBP	76.59 ± 10.78	77.56 ± 10.26	0.2184
Total Cholesterol	187.51 ± 37.23	194.62 ± 36.20	0.011
HDL-cholesterol	46.84 ± 12.23	48.83 ± 11.78	0.029
FEV1 (%)	83.81 ± 17.45	92.25 ± 13.33	<0.0001
FVC (%)	88.97 ± 13.02	92.71 ± 11.80	<0.0001
FEV1/FVC	0.70 ± 0.12	0.78 ± 0.07	<0.0001
AST *	23.24 (3.09–24.57)	21.84 (3.08–21.93)	0.0131
ALT *	19.54 (2.89–21.16)	19.35 (2.96–19.48)	0.8
TG *	116.64 (4.68–126.02)	121.39 (4.79–122.36)	0.3731

Values are number (%) or mean ± standard deviation. * Geographic mean (95% confidence interval). Abbreviation: ALT, alanine aminotransferase; AST, aspartate aminotransferase; BMI, body mass index; DBP, diastolic blood pressure; FEV1, forced expiratory volume in one second; FVC, forced vital capacity; HDL, high-density lipoprotein; SBP, systolic blood pressure; TG, triglycerides.

**Table 3 jcm-11-01077-t003:** Incidence and risk of lung cancer according to lung function patterns.

Lung Function	Total No. (n)	Lung Cancer Cases (n)	Lung Cancer Incidence (Per 1000 Person-Years)	HR (95% CI)		
				Model 1	Model 2	Model 3
Normal	15,775	78	0.76	1 (reference)	1 (reference)	1 (reference)
Obstructive	2686	72	4.38	5.817 (4.222–8.014)	2.518 (1.739–3.648)	2.505 (1.729–3.629)
Non-obstructive	2092	19	1.40	1.837 (1.112–3.033)	1.296 (0.776–2.166)	1.273 (0.761–2.129)

Abbreviation: CI, confidential interval; HR, hazard ratio. Non-adjustment for Model 1; Adjustments for Model 2: age, sex, BMI, income, smoking, alcohol consumption and moderate physical activity; Adjustments for Model 3: Model 2 plus hypertension and hypercholesterolemia.

**Table 4 jcm-11-01077-t004:** Incidence and risk of lung cancer according to lung function quartiles.

Lung Function	Total No. (*n*)	Lung Cancer Cases (*n*)	Lung Cancer Incidence (Per 1000 Person-Years)	HR (95% CI)		
				Model 1	Model 2	Model 3
FEV1 *						
Q1 (highest)	5138	23	0.69	1 (reference)	1 (reference)	1 (reference)
Q2	5139	27	0.81	1.188 (0.681–2.071)	1.442 (0.823–2.526)	1.444 (0.824–2.529)
Q3	5138	40	1.21	1.769 (1.059–2.955)	1.960 (1.167–3.292)	1.965 (1.169–3.302)
Q4 (lowest)	5138	79	2.41	3.528 (2.217–5.613)	2.854 (1.776–4.589)	2.845 (1.769–4.575)
FVC ^#^						
Q1	5138	32	0.95	1 (reference)	1 (reference)	1 (reference)
Q2	5139	37	1.11	1.171 (0.730–1.88)	1.231 (0.766–1.979)	1.234 (0.767–1.983)
Q3	5138	34	1.03	1.085 (0.670–1.759)	1.026 (0.631–1.670)	1.038 (0.638–1.690)
Q4	5138	66	2.03	2.141 (1.403–3.265)	1.513 (0.978–2.340)	1.507 (0.973–2.333)
FEV1/FVC *						
Q1	5138	14	0.41	1 (reference)	1 (reference)	1 (reference)
Q2	5139	20	0.60	1.483 (0.749–2.936)	1.149 (0.579–2.282)	1.148 (0.578–2.280)
Q3	5166	37	1.12	2.756 (1.490–5.097)	1.614 (0.861–3.023)	1.619 (0.864–3.034)
Q4	5110	98	3.09	7.696 (4.395–13.476)	2.886 (1.571–5.301)	2.891 (1.573–5.312)

Abbreviation: CI, confidential interval; FEV1, forced expiratory volume in one second; FVC, forced vital capacity; HR, hazard ratio. Non-adjustment for Model 1; adjustments for Model 2: age, sex, BMI, income, smoking, alcohol consumption and moderate physical activity; Adjustments for Model 3: Model 2 plus hypertension and hypercholesterolemia. * *p* for trend < 0.05 for all models, ^#^
*p* for trend < 0.05 for model 1.

**Table 5 jcm-11-01077-t005:** Incidence and risk of lung cancer according to lung function deciles.

Lung Function	Total No. (*n*)	Lung Cancer Cases (*n*)	Lung Cancer Incidence (Per 1000 Person-Years)	HR (95% CI)		
				Model 1	Model 2	Model 3
FEV1 *						
D1 (highest)	2055	11	0.82	1 (reference)	1 (reference)	1 (reference)
D2	2055	11	0.81	0.984 (0.426–2.269)	1.265 (0.546–2.928)	1.265 (0.546–2.929)
D3	2056	7	0.53	0.643 (0.249–1.658)	0.876 (0.338–2.270)	0.887 (0.342–2.300)
D4	2055	13	0.97	1.180 (0.529–2.634)	1.600 (0.712–3.594)	1.604 (0.714–3.605)
D5	2056	8	0.60	0.732 (0.295–1.820)	1.053 (0.420–2.640)	1.058 (0.422–2.653)
D6	2055	15	1.13	1.373 (0.630–2.989)	1.833 (0.835–4.023)	1.838 (0.837–4.037)
D7	2055	12	0.91	1.099 (0.485–2.490)	1.400 (0.612–3.199)	1.413 (0.618–3.230)
D8	2056	27	2.08	2.526 (1.253–5.092)	2.955 (1.448–6.031)	2.964 (1.452–6.050)
D9	2055	20	1.54	1.876 (0.899–3.915)	1.944 (0.921–4.101)	1.931 (0.915–4.077)
D10 (lowest)	2055	45	3.39	4.118 (2.130–7.961)	3.269 (1.670–6.397)	3.277 (1.674–6.416)
FVC ^#^						
D1	2055	11	0.82	1 (reference)	1 (reference)	1 (reference)
D2	2055	14	1.03	1.254 (0.569–2.763)	1.279 (0.580–2.821)	1.269 (0.576–2.799)
D3	2056	18	1.34	1.627 (0.768–3.445)	1.707 (0.805–3.622)	1.701 (0.802–3.608)
D4	2055	12	0.90	1.098 (0.485–2.489)	1.169 (0.515–2.657)	1.165 (0.513–2.647)
D5	2056	14	1.06	1.292 (0.587–2.846)	1.352 (0.612–2.988)	1.349 (0.610–2.982)
D6	2055	11	0.83	1.017 (0.441–2.345)	1.008 (0.435–2.333)	1.007 (0.435–2.333)
D7	2055	20	1.50	1.831 (0.877–3.822)	1.731 (0.825–3.632)	1.748 (0.833–3.669)
D8	2056	13	0.99	1.205 (0.540–2.691)	1.074 (0.479–2.408)	1.079 (0.481–2.424)
D9	2055	24	1.88	2.303 (1.128–4.701)	1.826 (0.886–3.764)	1.803 (0.874–3.721)
D10	2055	32	2.43	2.961 (1.493–5.875)	1.792 (0.889–3.613)	1.773 (0.879–3.577)
FEV1/FVC *						
D1	2078	6	0.43	1 (reference)	1 (reference)	1 (reference)
D2	2028	5	0.37	0.875 (0.267–2.869)	0.764 (0.233–2.506)	0.767 (0.234–2.515)
D3	2096	8	0.58	1.356 (0.470–3.908)	1.099 (0.381–3.173)	1.097 (0.380–3.168)
D4	2079	7	0.52	1.226 (0.412–3.649)	0.902 (0.302–2.694)	0.905 (0.303–2.703)
D5	1996	8	0.62	1.476 (0.512–4.253)	1.003 (0.346–2.905)	1.007 (0.347–2.917)
D6	2069	16	1.18	2.787 (1.090–7.122)	1.692 (0.656–4.363)	1.705 (0.661–4.400)
D7	2053	11	0.85	2.013 (0.744–5.443)	1.111 (0.406–3.041)	1.107 (0.404–3.032)
D8	2026	17	1.32	3.129 (1.234–7.937)	1.495 (0.577–3.876)	1.516 (0.585–3.932)
D9	2064	29	2.25	5.339 (2.216–12.862)	2.212 (0.890–5.496)	2.215 (0.891–5.508)
D10	2064	62	4.92	11.742 (5.078–27.151)	3.579 (1.462–8.761)	3.593 (1.467–8.801)

Abbreviation: CI, confidential interval; FEV1, forced expiratory volume in one second; FVC, forced vital capacity; HR, hazard ratio. Non-adjustment for Model 1; Adjustments for Model 2: age, sex, BMI, income, smoking, alcohol consumption and moderate physical activity; Adjustments for Model 3: Model 2 plus hypertension and hypercholesterolemia. * *p* for trend < 0.05 for all models, ^#^
*p* for trend < 0.05 for model 1.

**Table 6 jcm-11-01077-t006:** Risk of lung cancer according to lung-function-based continuous variables.

Continuous Variables	Model 1		Model 2		Model 3	
	HR (95% CI)	*p* Value	HR (95% CI)	*p* Value	HR (95% CI)	*p* Value
FVC per 1%	0.973 (0.961–0.985)	<0.0001	0.988 (0.976–1.000)	0.0527	0.988 (0.976–1.000)	0.0565
FEV1 per 1%	0.960 (0.952–0.969)	<0.0001	0.974 (0.965–0.983)	<0.0001	0.974 (0.965–0.983)	<0.0001
FEV1/FVC per 0.01	0.922 (0.911–0.932)	<0.0001	0.950 (0.936–0.965)	<0.0001	0.950 (0.935–0.964)	<0.0001

Abbreviation: CI, confidential interval; FEV1, forced expiratory volume in one second; FVC, forced vital capacity; HR, hazard ratio. Non-adjustment for Model 1; adjustments for Model 2: age, sex, BMI, income, smoking, alcohol consumption and moderate physical activity; adjustments for Model 3: Model 2 plus hypertension and hypercholesterolemia.

**Table 7 jcm-11-01077-t007:** Incidence and risk of lung cancer according to combination of FEV1 and FVC quartile.

FEV_1_	FVC	Total No. (*n*)	Lung Cancer Cases (*n*)	Lung Cancer Incidence (Per 1000 Person-Years)	HR (95% CI)		
					Model 1	Model 2	Model 3
Q1–3	Q1–3	13871	81	0.90	1 (reference)	1 (reference)	1 (reference)
Q1–3	Q4	1544	9	0.92	1.023 (0.514–2.037)	0.677 (0.337–1.362)	0.672 (0.334–1.351)
Q4	Q1–3	1544	22	2.20	2.448 (1.528–3.921)	1.755 (1.083–2.845)	1.754 (1.084–2.847)
Q4	Q4	3594	57	2.50	2.790 (1.988–3.915)	1.907 (1.345–2.703)	1.889 (1.331–2.681)

Q1–3 = higher three quartiles; Q4 = lowest quartile. Abbreviation: CI = confidential interval; FEV1 = forced expiratory volume in one second; FVC = forced vital capacity; HR = hazard ratio. Non-adjustment for Model 1; adjustments for Model 2: age, sex, BMI, income, smoking, alcohol consumption and moderate physical activity; adjustments for Model 3: Model 2 plus hypertension and hypercholesterolemia.

## Data Availability

Publicly available datasets were analyzed in this study. This data can be found here: KCDC (http://knhanes.cdc.go.kr/knhanes/eng/index.do, accessed on 16 December 2021) and NHIS sharing service websites (http://nhiss.nhis.or.kr, accessed on 16 December 2021).

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
