# Peer review of "Impaired Lung Function and Lung Cancer Incidence: A Nationwide Population-Based Cohort Study"

_jcm, 2022, doi:10.3390/jcm11041077_

Round 1
Reviewer 1 Report
Re: Impaired Lung Function and Lung Cancer Incidence: A Nationwide Population-Based Cohort Study
This article reports the relationship between lung function and lung cancer incidence. It is a large cohort study and may help anything in the future. However, I think the conclusions are weak. Besides your conclusions, there are some problems in the following points.
・Table 5 shows the incidence of lung cancer in lung function deciles. However, in Model 2 and 3, only a small part (D8 and D10) shows the difference from the reference (D1). This result is an overestimate to conclude that lower FEV1 or FEV1/FVC value may be associated with increased risk of lung cancer. Further studies in model 1-3 should be made using the continuous variables of FEV1 and FEV1/FVC value.
・The word "minimal lung function impairment" is often used in your article, what is its definition? Is it better lung function group or worse lung function group? You had better specify the definition clearly.
Author Response
Dear Editor-in-Chief,
We would like to thank you and the reviewers of the Journal of Clinical Medicine for taking the time to review our article.
We have made some corrections and clarifications in the manuscript after going over the reviewers’ comments.
The changes are summarized below:
- Table 5 shows the incidence of lung cancer in lung function deciles. However, in Model 2 and 3, only a small part (D8 and D10) shows the difference from the reference (D1). This result is an overestimate to conclude that lower FEV1 or FEV1/FVC value may be associated with increased risk of lung cancer. Further studies in model 1-3 should be made using the continuous variables of FEV1 and FEV1/FVC value.
Response : We agreed with reviewer’s opinion. We further analyzed about the risk of lung cancer according to lung function based on continuous variables. We also identified the lung cancer risk according to FEV1 and FEV1/FVC ratio decline based on continuous value. We added the table 6 and related comments at 3.5. Risk of lung cancer according to continuous variables of lung function.
The word "minimal lung function impairment" is often used in your article, what is its definition? Is it better lung function group or worse lung function group? You had better specify the definition clearly.
Response : Thank you for your comments. We intended to identify the lung cancer risk according to minimal lung function impairment, not definite better or worse lung function group. So we classify participants according to deciles or increase of 1% or 0.05 in continuous variables as the reviewer suggested.

Reviewer 2 Report
Thank you for giving me the opportunity to review it. The paper discusses the risk of low lung function and lung cancer, but I think it would be more clear if you directly discussed the causal relationship between smoking-related and lung cancer. It is generally easier to discuss the association between smoking and lung cancer than to go to the trouble of discussing the relationship between respiratory function and lung cancer. I judge that it would be preferable to change the discussion to directly discuss the causal relationship with smoking.
Author Response
Dear Editor-in-Chief,
We would like to thank you and the reviewers of the Journal of Clinical Medicine for taking the time to review our article.
We have made some corrections and clarifications in the manuscript after going over the reviewers’ comments.
The changes are summarized below:
Reviewer 2:
1. The paper discusses the risk of low lung function and lung cancer, but I think it would be more clear if you directly discussed the causal relationship between smoking-related and lung cancer. It is generally easier to discuss the association between smoking and lung cancer than to go to the trouble of discussing the relationship between respiratory function and lung cancer. I judge that it would be preferable to change the discussion to directly discuss the causal relationship with smoking.
Resposne : We agreed the reviewer’s opinion. It is obvious that Smoking is one of definite risk factor for lung cancer. However, the proportions of non-smoking lung cancer patients are increasing. The reasons for this phenomenon are genetic variables, other environment factors such as pollutants and underlying chronic inflammatory lung disease. We focused on lung function and incidence of lung cancer, and we showed that minimal lung function impairment was one of independent risk factor for lung cancer development after adjusting confounding factors including smoking. We added detailed contents at paragraph 6 of discussion section.

Reviewer 3 Report
Dear Authors and Editors,
thank you very much for presenting this study to me for review. The issue of the relationship between the reduction of lung function parameters and the risk of developing lung cancer is very interesting. The results of this study may have significant application in clinical practice.
The study is very well conducted, the presentation of the goals is clear, the results and conclusions are interesting.
Below you will find some of my suggestions:
- The only question that seems not clear enough to me: what was the purpose of assessing the activity of transaminases, the results of which can be found in Tables 1 and 2? Were they part of the patient's overall laboratory assessment?
Lines: 44-46: In addition, chronic obstructive pulmonary disease (COPD), a smoking-related group of conditions that block airflow and make it difficult to breathe has been found to be associated with a higher rate of lung cancer in several studies [5-8].
I would change this sentence: In addition chronic obstructive pulmonary disease (COPD) and other smoking related conditions has been found to be associated with higher rate of lung cancer in several studies [5-8].
In conclusion I have to state that, in my opinion, clinicians and their patients can benefit greatly from sharing the results of this study with them.
Best regards.
Author Response
Dear Editor-in-Chief,
We would like to thank you and the reviewers of the Journal of Clinical Medicine for taking the time to review our article.
We have made some corrections and clarifications in the manuscript after going over the reviewers’ comments.
The changes are summarized below:
Reviewer 3:
1. The only question that seems not clear enough to me: what was the purpose of assessing the activity of transaminases, the results of which can be found in Tables 1 and 2? Were they part of the patient's overall laboratory assessment?
Response : AST and ALT were the part of the patients’ overall laboratory assessment. If it doesn’t fit the whole topic, we will delete it.
2. Lines: 44-46: In addition, chronic obstructive pulmonary disease (COPD), a smoking-related group of conditions that block airflow and make it difficult to breathe has been found to be associated with a higher rate of lung cancer in several studies. I would change this sentence: In addition chronic obstructive pulmonary disease (COPD) and other smoking related conditions has been found to be associated with higher rate of lung cancer in several studies.
Response: As the reviewer suggested, we corrected the sentence in line 44-46.

Round 2
Reviewer 1 Report
It has been corrected for the pointed out parts.
From the results of additional continuous variables, I think the conclusions drawn from the results of the D1-10 classification are by no means overestimated.
Statistically, I think newer results are consistent with this journal's conclusions.
Author Response
- It has been corrected for the pointed out parts.
From the results of additional continuous variables, I think the conclusions drawn from the results of the D1-10 classification are by no means overestimated.
Statistically, I think newer results are consistent with this journal's conclusions.
Response : Thanks to the reviewer, the conclusion of the paper has more evidence.

Reviewer 2 Report
Thank you for giving me the opportunity to write this paper. It is interesting, but as other reviewers have pointed out, I think the content needs to be substantially reorganized. I would like to request a major reconsideration as a rejection at first.
Author Response
1. Thank you for giving me the opportunity to write this paper. It is interesting, but as other reviewers have pointed out, I think the content needs to be substantially reorganized. I would like to request a major reconsideration as a rejection at first.
Response : Thank you for your considerable review our article. As the reviewer pointed out, our article had many limitations. However, this study showed that minimal lung function impairment could be attribute to the risk of lung cancer in general populations even after adjusting for confounding variables, and would be helpful as a basic data for the clinical application of pulmonary function tests, which is easy and non-invasive, in lung cancer screening.
Round 3
Reviewer 2 Report
As the authors state, this study showed that even after adjusting for confounding variables, minimal lung function impairment in the general population is attributable to lung cancer risk, and may be useful as basic data for the clinical application of simple, non-invasive lung function tests for lung cancer screening. However, overall, it is desirable to make sure that the methods are well described and to accept the manuscript after checking the English proofreading.
Author Response
Thank you for your considerable review our article. As the reviewer pointed out, we further check the English proofreading. We attach the certification of English proofreading and revised manuscript.
